# SimpleMind: An open-source software environment that adds thinking to deep neural networks

**Youngwon Choi, M. Wasil Wahi-Anwar, Matthew S. Brown** *

Center for Computer Vision and Imaging Biomarkers, Department of Radiological Sciences, David Geffen School of Medicine at UCLA, University of California, Los Angeles, Los Angeles, CA, United States of America

* mbrown@mednet.ucla.edu

## Abstract

Deep neural networks (DNNs) detect patterns in data and have shown versatility and strong performance in many computer vision applications. However, DNNs alone are susceptible to obvious mistakes that violate simple, common sense concepts and are limited in their ability to use explicit knowledge to guide their search and decision making. While overall DNN performance metrics may be good, these obvious errors, coupled with a lack of explainability, have prevented widespread adoption for crucial tasks such as medical image analysis. The purpose of this paper is to introduce *SimpleMind*, an open-source software environment for Cognitive AI focused on medical image understanding. It allows creation of a knowledge base that describes expected characteristics and relationships between image objects in an intuitive human-readable form. The knowledge base can then be applied to an input image to recognize and understand its content. SimpleMind brings thinking to DNNs by: (1) providing methods for reasoning with the knowledge base about image content, such as spatial inferencing and conditional reasoning to check DNN outputs; (2) applying process knowledge, in the form of general-purpose software agents, that are dynamically chained together to accomplish image preprocessing, DNN prediction, and result post-processing, and (3) performing automatic co-optimization of all knowledge base parameters to adapt agents to specific problems. SimpleMind enables reasoning on multiple detected objects to ensure consistency, providing cross-checking between DNN outputs. This machine reasoning improves the reliability and trustworthiness of DNNs through an interpretable model and explainable decisions. Proof-of-principle example applications are provided that demonstrate how SimpleMind supports and improves deep neural networks by embedding them within a Cognitive AI environment.

## Introduction

Deep neural networks (DNNs) detect patterns in data and have shown versatility and strong performance in many computer vision applications. However, despite many research

---

**Data Availability Statement:** The open source software is available on GitLab: https://gitlab.com/sm-ai-team/simplemind. Public data used for the prostate application is available at: https://promise12.grand-challenge.org. Public data used

for the kidney application is available at: https://kits19.grand-challenge.org/data/. "Portions of the data used for kidney and chest x-ray applications are collected from UCLA and restricted for sharing because of the sensitive patient information. This restriction is imposed by Radiological Sciences at UCLA. Dr. Mark DeMars (Director of Research, UCLA Radiological Sciences, mdemars@mednet.ucla.edu) is a non-author, institutional point of contact.

**Funding:** The authors received no specific funding for this work.

**Competing interests:** The authors have declared that no competing interests exist.

publications there has not been broad adoption of artificial intelligence (AI) in crucial tasks such as medical imaging. Medical researchers have pointed to potential overestimation of DNN performance [1], diminished DNN performance on external datasets [2], lack of consistent performance across cohorts [3], and the need to build trustworthy AI [4].

DNNs can be considered a "stimulus-response" function without the thinking, using knowledge and reasoning, that makes human vision superior. DNNs are susceptible to obvious ("dumb") mistakes that violate simple common sense concepts and are limited in their ability to use explicit knowledge to guide their search and decision making. While overall DNN performance metrics may be good, these obvious errors, coupled with a lack of explainability, have prevented widespread adoption for crucial tasks such as medical image analysis.

To improve computer vision accuracy and reliability we embed deep neural networks within a Cognitive AI environment to meld pattern recognition with conceptual knowledge and reasoning. Cognitive AI includes not only the learning of patterns in data, but also learning through teaching and concepts (declared knowledge) as well as reasoning to apply this knowledge to guide the interpretation of a specific image. We provide an implementation of "thinking" that encompasses both learning and reasoning.

The purpose of this paper is to introduce *SimpleMind*, an open-source software environment for image understanding, i.e., segmenting and recognizing image elements to form a coherent high-level model of a scene where reasoning can be performed. The SimpleMind environment brings thinking to DNNs by:

1. allowing creation of a knowledge base for a type of image,

2. providing methods for reasoning with the knowledge base about image content such as spatial inferencing and conditional reasoning to cross check multiple neural network outputs,

3. applying process knowledge, in the form of general-purpose software agents, that are dynamically chained together to accomplish image preprocessing, DNN prediction, and result post-processing, and

4. performing automatic co-optimization of all knowledge base parameters to adapt agents to specific problems.

SimpleMind uses a multi-agent architecture as a runtime environment for processing (understanding) an image using a knowledge base. The knowledge base is decoupled from the processing and uses a semantic network to represent domain knowledge intuitively and to guide agents that operate in the environment—for example, segmentation agents that include DNN agents and machine reasoning agents that evaluate the results. Within the environment, users can develop a SimpleMind application by creating a knowledge base that is applied to an input image to recognize its medical elements. A strength of this Cognitive AI environment is that it enables two types of user interactions:

1. it allows non-programmer end-users to build a medical application directly and completely using their domain knowledge without knowing the details of the processing code;

2. it provides an open environment for algorithm developers to add or extend processing agents and thus contribute generally without being tied to a particular medical problem.

We will describe the SimpleMind environment for Cognitive AI and demonstrate its flexible application to medical image segmentation on chest x-ray [5], kidney CT [6, 7], and prostate MRI [8].

## Knowledge representation

SimpleMind adds reasoning to deep neural networks using a knowledge base that is explicit, human-readable, and decoupled from processing algorithms. This stands in contrast to implicit ("black box") knowledge encoded within DNN weights or in pre/post processing code. In a SimpleMind application for a particular type of image, the knowledge base defines what is known and can be considered "long term memory". The representation is based on semantic networks, originally developed and applied in language processing [9], and is intuitive and allows easy creation, extension, and re-use of knowledge.

The knowledge base for a SimpleMind application is created as a semantic network (SN) where each node represents a concept—typically an object, object component, or object state. The SimpleMind environment provides a human-readable intuitive language to specify a semantic network. Each node contains attributes that describe expected object characteristics relating to size, shape, pixel intensity, and relative position. Spatial relationships that can be described between objects, include part of, right of, left of, above, below, inside, etc. Attributes are derived from a vocabulary that defines the name of the attribute and its associated parameters. For example, the vocabulary defines "PartOf" as physical composition with a single parameter, the name of the related parent node. When used to create an attribute in a semantic network, "PartOf" forms a relational link between two nodes. The vocabulary also defines attributes with numerical parameters, such as "RightOf", which includes two parameters: (1) the related node, and (2) the expected distance to the right. Fuzzy sets are currently used to represent prior expectations for object characteristics using a confidence function over the range of possible parameter values [10]. The fuzzy functions can be set initially by a human expert and refined by learning from data (see Section Learning and Optimization).

In addition to content knowledge (e.g., anatomical knowledge in medical images), the semantic network attributes can represent procedural knowledge used by processing agents, including DNN architectures (e.g., U-Net, ResNet, or any user-defined architectures), DNN weight optimization algorithms (e.g., Adam optimizer), learning hyper parameters, and image pre and post processing parameters. Crucially, all attribute parameters in the semantic network are exposed (separate from the processing code) and human readable, so they can be both specified by a human and auto optimized by SimpleMind (see Section Learning and Optimization). The SimpleMind environment allows a DNN agent to train DNN weights using the above attribute parameters from a given SN node. The DNN weights are then stored with the node, embedding the DNN within the semantic network. Thus a SimpleMind knowledge base can include both declared knowledge (that it is "taught") and learned knowledge from examples (acquired through machine learning), i.e., we can actively teach the Cognitive AI as well as have it learn passively from data.

The environment provides tools for a user to create a knowledge base for their application by adding semantic network nodes and attributes. The nodes of a semantic network are stored in human readable text files. The network can be scaled using the existing vocabulary by adding nodes and attributes that expand the knowledge base. The SimpleMind vocabulary can be also extended, expanding the attributes that can be used when creating a knowledge base.

The semantic network provides a direct approach to capturing knowledge, as opposed to indirectly represented and entangled within processing code. With this abstraction, developing a SimpleMind application is like teaching or instructing a human at a cognitive level without writing code for each processing step. The SimpleMind environment handles the aggregation and chaining of many processing agents, enabling a large scale Cognitive AI system. In the next section we will describe how processing agents use the knowledge base.

## Multi-agent thinking

In SimpleMind, computer vision involves recognizing objects (nodes) from the knowledge base in a given image. Multiple software agents work together to segment the image into candidate regions, then select the best candidate based on object attributes described in the knowledge base. Software agents collaborate to solve the vision problem by reading from, and writing to, a global Blackboard data structure [11]. The analogy is a team of specialists observing and writing on a blackboard to contribute their specific expertise to solving a problem jointly. The Blackboard is the working space, or "short term memory", of SimpleMind during the "thinking" process, i.e., during comparison and matching of the image to a knowledge base for image understanding. An agent can read information from the Blackboard generated by other agents and add or update information. Agents operate independently and collaborate only via the Blackboard, giving them a degree of autonomy and making the system more flexible and scalable.

For each node in the semantic network, a data structure called a Solution Element is created on the Blackboard, corresponding to an object to be recognized in the image. The Solution Element stores all agent contributions while recognizing the object. Knowledge base attributes and candidate image regions are transformed into a common feature space for selection of the best candidate. A Knowledge Agent accesses the knowledge base and creates each Solution Element, initializing it with prior expectations for object feature values. Knowledge base attributes reflect commonly understood concepts, such as left, right, above, below; these spatial attributes are transformed to a "relative centroid position" feature with x, y, and z axis parameters that are set according to the particular attribute. Knowledge base object attributes can be transformed to unary features (e.g., size, shape, intensity) or binary relational features (e.g., spatial relationships). Thus, the objects and their relationships represented in the semantic network are transformed into a directed graph of Solution Elements on the Blackboard, with the direction of the link reflecting the dependency of an object's attribute upon another object.

Solution Elements are processed sequentially by agents. Objects are recognized in order based on the directed links between their Solution Elements. A Scheduling Agent determines the order dynamically—after one Solution Element is processed, the next is then determined. The Scheduling Agent selects the Solution Element with the highest percentage of features that can be computed, i.e., where prerequisite Solution Elements have already been processed. This approach is flexible and opportunistic, it can automatically handle any set of Solution Elements; and if a particular object is not present or not found in a particular image, the ordering can change without requiring hard-coded exception rules.

When a particular Solution Element is scheduled for processing, a Reasoning Agent computes an image search area using spatial inferencing from the relationships to previously recognized objects. This search area is provided as a mask to guide a Segmentation Agent that generates candidate image regions for the object. Segmentation is typically performed by a DNN agent that generates multiple connected components as candidate regions. Feature values are computed for each candidate region and compared against the expected values by a Reasoning Agent. Thus, by pattern classification, the candidate that best matches these expectations from the knowledge base is selected.

Agent types currently provided in SimpleMind and shown in Fig 1 include:

- Knowledge Agents: to transform the knowledge in the semantic network into Solution Elements on the Blackboard and attributes to feature space

- Scheduling Agents: to determine which Solution Element should be processed next, based on graph dependencies as described above

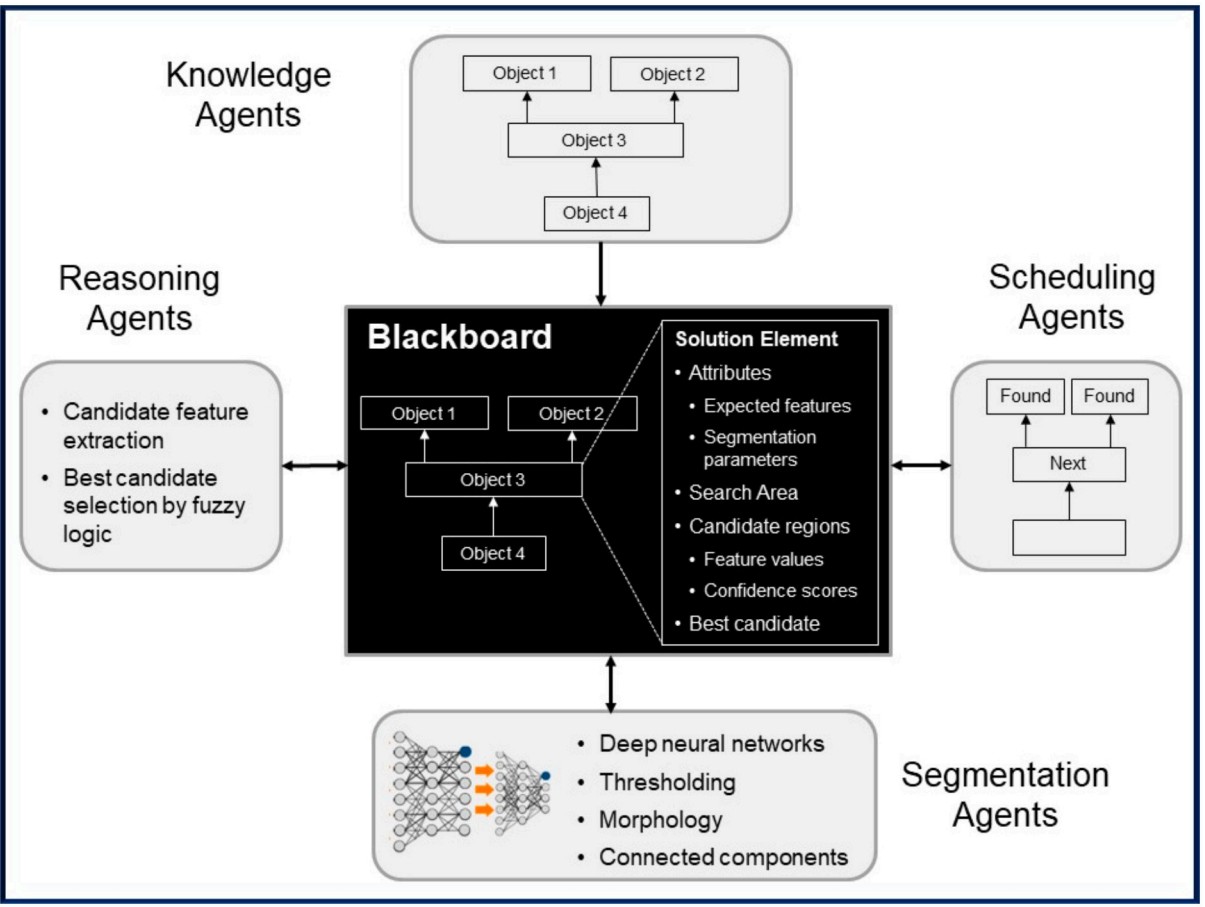

**Fig 1. SimpleMind multi-agent thinking architecture.** Schematic showing agent types and the Blackboard for sharing of results during image understanding.

- Segmentation Agents: including intensity thresholding, mathematical morphology, and deep convolutional neural networks

  - The DNN agent will perform training to learn the weights if they do not already exist

- Reasoning Agents: to (1) derive search areas for objects (to guide Segmentation Agents), and (2) evaluate candidate image regions and select the best match to an object based on features.

Agents are activated iteratively, one at a time. At each iteration an activation score is computed for each registered agent. The agent with the highest score is activated and can contribute to the solution on the Blackboard. Each agent provides a function to compute its activation score based on the contents of the Blackboard, in particular whether the Solution Element being processed has the relevant attributes and necessary data required by the agent. The process repeats until all activation scores are zero and no further agents activate. The system control is simple, yet highly flexible, with agent priorities determined through their activation functions.

The contents of the Blackboard reflect what SimpleMind is thinking at any point in time and its current understanding of the image. Once all Solution Elements have been processed,

the Blackboard contains an instantiation of the general knowledge base to a particular image. The object attributes from the general knowledge base are now instantiated with actual numerical feature values from the image, from the particular patient anatomy in the case of medical imaging, enabling further high-level reasoning.

## Application examples

As proof-of-principle, we provide three example SimpleMind applications, describing each knowledge base and how the agents make use of it. In particular, we describe how thinking using the knowledge base improves deep neural networks.

### Kidney segmentation on CT images

State of the art kidney segmentation on CT uses DNNs with ad hoc connected component post-processing. Although performance is good, there is an opportunity for improvement with increased reliability by applying anatomical knowledge using SimpleMind. This knowledge is used to: (1) improve discrimination of adjacent abdominal organs with similar image intensities, (2) select the best connected component based on size and relative location.

The knowledge base provides the following basic relevant information about the kidneys:

- The kidneys have a soft tissue image intensity range of 300–2000 Hounsfield Units (HU) on contrast-enhanced CT scans

- The kidneys do not overlap spatially with bone (they are adjacent and of similar intensity in contrast-enhanced CT scans)

- The right kidney is 5–10 cm to the right of the spine

- The left kidney is 5–10 cm to the left of the spine

- The right and left kidneys are at the same craniocaudal level in the abdomen

This kidney knowledge is reflected as attributes in the knowledge base; it includes relationships to the abdomen, bone, and spine. As such, these objects are also described as nodes in the semantic network (SN) shown in Fig 2. These related nodes surround, and provide context to, a DNN node for a data-driven kidney segmentation. This allows the knowledge base to review, select, fine-tune the segmentation candidates predicted from a conventional DNN, three-dimensional U-Net [12].

The SimpleMind SN is stored as human readable text files, with a "Node List" file being the entry point that lists the nodes in the SN. The attributes of each node are then stored in a text file. Fig 2 shows the SimpleMind SN for kidney CT segmentation. This kidney SN Node List file is shown in Fig 2A. Fig 2B–2D are example node files that contain the attributes describing general knowledge about an object. For relational attributes, the node name of the related object is given. A visualization of the relational links between nodes in the kidney SN is shown in Fig 2E.

The abdomen node in Fig 2B describes the attributes of the abdomen as imaged on CT:

- Line 2 gives the CT HU intensity range of [-400, 2000] (which can guide intensity thresholding)

- Line 5 gives the average cross-sectional area of the abdomen tissue within the HU threshold range; since this varies between patients, it is represented using a fuzzy membership function that assigns a confidence [0.0, 1.0] to each area value; the membership function is piecewise

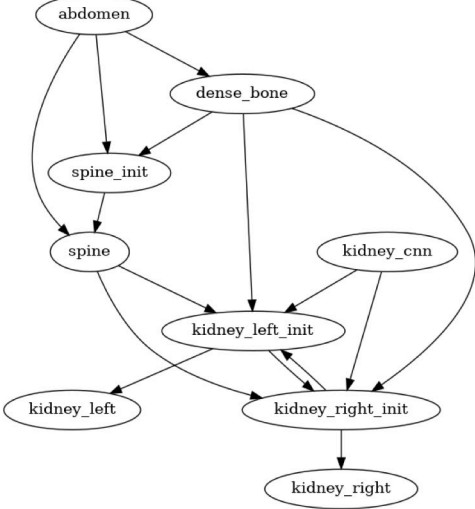

(A) Node List file of the semantic network for kidney (kidney SN)

```
Model: kidney_model
1
kidney_right
kidney_right_init
kidney_left
kidney_left_init
kidney_cnn
spine
spine_init
dense_bone
abdomen
End: kidney_model;
```

(B) "abomen" node file

```
AnatPathEntity: abdomen;
HUrange -400 to 2000;
UseSubsampledImage 4 4 2;
DoNotFormCandsWith_NumVoxelsLessThan 1000;
Area_PerXYplane_cm2 [(100, 0) (500, 1)];
End: abdomen;
```

(C) "kidney_left_init" node file

```
AnatPathEntity: kidney_left_init;
DoNotFormCandsWith_NumVoxelsLessThan 500;
PartOf_E kidney_cnn;
NotPartOf dense_bone;
LeftOf_PlanarCentroid_mm spine [(0,0) (50,1) (100,1) (150,0)];
NearZ_Centroid_mm kidney_right_init [(-100,0) (-30,1) (30,1) (100,0)];
Volume_cc [(0,0) (150,1)];
End: kidney_left_init;
```

(D) "kidney_right_init" node file

```
AnatPathEntity: kidney_right_init;
DoNotFormCandsWith_NumVoxelsLessThan 500;
PartOf_E kidney_cnn;
NotPartOf dense_bone;
RightOf_PlanarCentroid_mm spine [(0,0) (50,1) (100,1) (150,0)];
NearZ_Centroid_mm kidney_left_init [(-100,0) (-30,1) (30,1) (100,0)];
Volume_cc [(0,0) (150,1)];
End: kidney_right_init;
```

(E) The relational links between nodes in the kidney SN

**Fig 2. SimpleMind semantic network (SN) for kidney segmentation.** The human readable text files of the SimpleMind semantic network (SN) for kidney segmentation. (A) The "Node List" text file of the SN where each line corresponds to a node in the SN. (B) The text file for the thorax node, containing the attributes for recognizing the thorax object. (C) The text file for the *kidney_left_init* node. (D) The text file for the *kidney_right_init*. (E) Graph visualization of the kidney SN.

linear with vertices specified. In this case, the confidence is 0 at 100 cm$^2$ or less, with a linear progression towards a full confidence of 1.0 at 500 cm$^2$ or more.

*dense_bone* and *spine* nodes have similar attributes describing their image intensity range and cross sectional area. The *kidney_cnn* node invokes a deep convolutional network to provide an initial segmentation that generates candidates for both the left and right kidneys; some of these candidates are expected to be the actual kidneys but may also include detections of incorrect stray regions.

The *kidney_left_init* node demonstrates the use of relational attributes to reliably identify the left kidney among the DNN connected components:

- Line 3 indicates that it is part of the `kidney_cnn` output

- Line 4 that it is not part of bone (the DNN occasionally includes bone within its initial segmentation since they have similar intensity in contrast-enhanced CT scans)

- Line 5 that it is left of the spine (typically 50–100 mm)

- Line 6 that it is at the same level anatomically as the right kidney in the longitudinal or z direction (typically within 3 cm of each other)

Machine reasoning is applied using these attributes to ensure that a candidate is selected that is in the appropriate location relative to the spine and right kidney. Following segmentation of `left_kidney_init`, there is a dependent node, `kidney_left`, that includes attributes relating to contiguity and smoothness that trigger hole filling and morphological smoothing operations to achieve the final result.

To illustrate how the SimpleMind kidney application can support a classical single DNN model approach, we chose a 3D U-Net architecture as a baseline, which is the same DNN model used in `kidney_cnn` node. The DNN is trained with the 210 images from the public KiTS19 (The 2019 Kidney and Kidney Tumor Segmentation Challenge) dataset [13] and 55 images from in-house UCLA kidney segmentation data collected between 2010 to 2022. The KiTS19 dataset is available in a public repository (https://kits19.grand-challenge.org/data/). The UCLA data can only be shared with permission from the institution. All training process parameters (such as batch size, learning rate) are written in human-readable form in the kidney application knowledge base (https://gitlab.com/sm-ai-team/simplemind-applications/-/blob/develop/ct_kidney_quick_start/kidney_sn/kidney_cnn). Fig 3 shows results from the SimpleMind kidney application and the baseline DNN model alone. The output from the baseline DNN alone (which is the same output from the `kidney_cnn` node) in Fig 3C shows an obvious (dumb) mistake with the DNN including an additional region that is

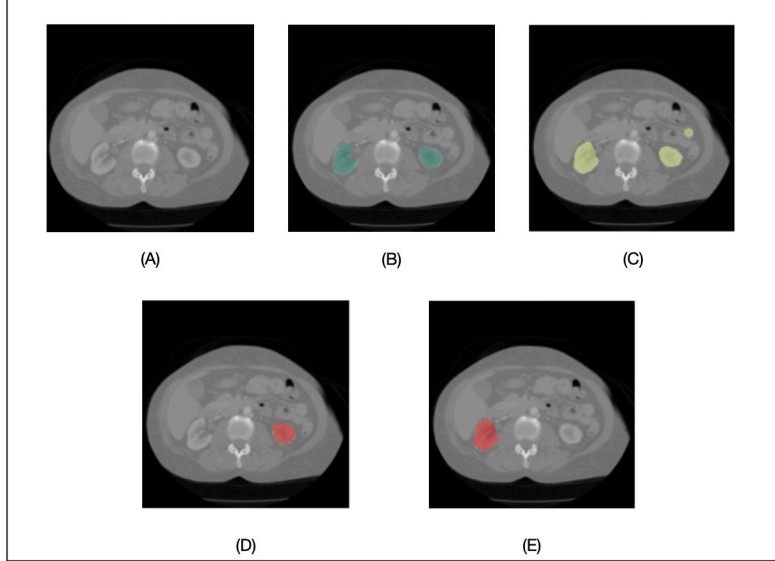

**Fig 3. Kidney segmentation on CT.** Kidney segmentation on CT: (A) original image; (B) reference kidney segmentation (green overlay); (C) segmentation result for the `kidney_cnn` node using a DNN alone (the baseline results); (D) SimpleMind segmentation of the `right_kidney` node (red overlay); (E) SimpleMind segmentation of the `left_kidney` node (red overlay).

disconnected from the kidneys. By applying the anatomical knowledge described above with machine reasoning, SimpleMind is able to separately identify the right and left kidneys in the *kidney_right* (Fig 3D) and *kidney_left* (Fig 3E) nodes, respectively, while excluding the extraneous region initially segmented by the DNN. This demonstrates the benefit of adding reasoning to improve DNN reliability. In experimental testing, the application achieved a Dice Coefficient for kidney segmentation of 0.881±0.145, and a Hausdorff distance (HD) metric of 46.4±55.8 mm [7]. In comparison, the baseline DNN trained in the same manner (without machine reasoning or anatomical knowledge) performed with a Dice coefficient of 0.878±0.144% and HD of 30.7±17.6 mm. While the Dice coefficient was almost unchanged, there was notable improvement in HD, in both its mean value and standard deviation. In our context, HD can be understood as a relative measure for "stray" segmented regions (i.e., DNN segmented regions that are separate from the target anatomy of interest), these can erode user trust if detected in unexpected or unreasonable anatomical locations and are time consuming to detect and edit by hand. The substantial improvement (reduction) in HD error suggests that stray segmented regions are excluded after applying machine reasoning to the DNN output within SimpleMind.

## Prostate segmentation in 3D MRI

Prostate segmentation in 3D T2-weighted MRI can be accomplished using a DNN, with performance being significantly influenced by application-specific preprocessing algorithms with data scientist hand-tuning. There is room for improvement by incorporating human heuristic knowledge using SimpleMind to overcome: 1) the relatively low contrast in the apex/base region of the prostate gland compared to the central region, 2) the presence of an endorectal coil in some patients that causes shape distortions in the prostate base and significant intensity histogram changes.

We built the SimpleMind prostate knowledge base to combine multiple DNNs for segmenting the center, apex, and base regions of the prostate and adding thinking to each DNN. The knowledge base mimics the following human expert behavior to compensate for the segmentation challenges described above:

- Experts adjust the window/level settings (intensity remapping) differently when viewing the apex and base regions to improve the perceived contrast based between prostate and background.

- Experts infer the shape of unclear or incomplete boundaries by considering adjacent slices and knowledge of the overall shape of the prostate.

This processing based on human expert knowledge is reflected in attributes in the prostate semantic network (prostate SN) shown in Fig 4. The prostate SN comprises of the Node List file in Fig 4A–4D are example node files for the image objects for related to the prostate apex region. The *prostate_apex_focused_cnn* node in Fig 4C includes attributes representing the learning hyper parameters of a DNN. Fig 4B and 4D show nodes that are used to apply thinking to the input and output of the DNN node *prostate_apex_focused_cnn*. *prostate_apex_box* specifies the image subregion that provides the histogram for normalization of the DNN input image. *prostate_apex_attention* refines and cross-checks the DNN output. A visualization of the relational links between nodes in the prostate SN is shown in Fig 4E. The knowledge base employs multiple DNN nodes to mimic the human behaviors in applying specific processing to the apex and base regions of the prostate gland.

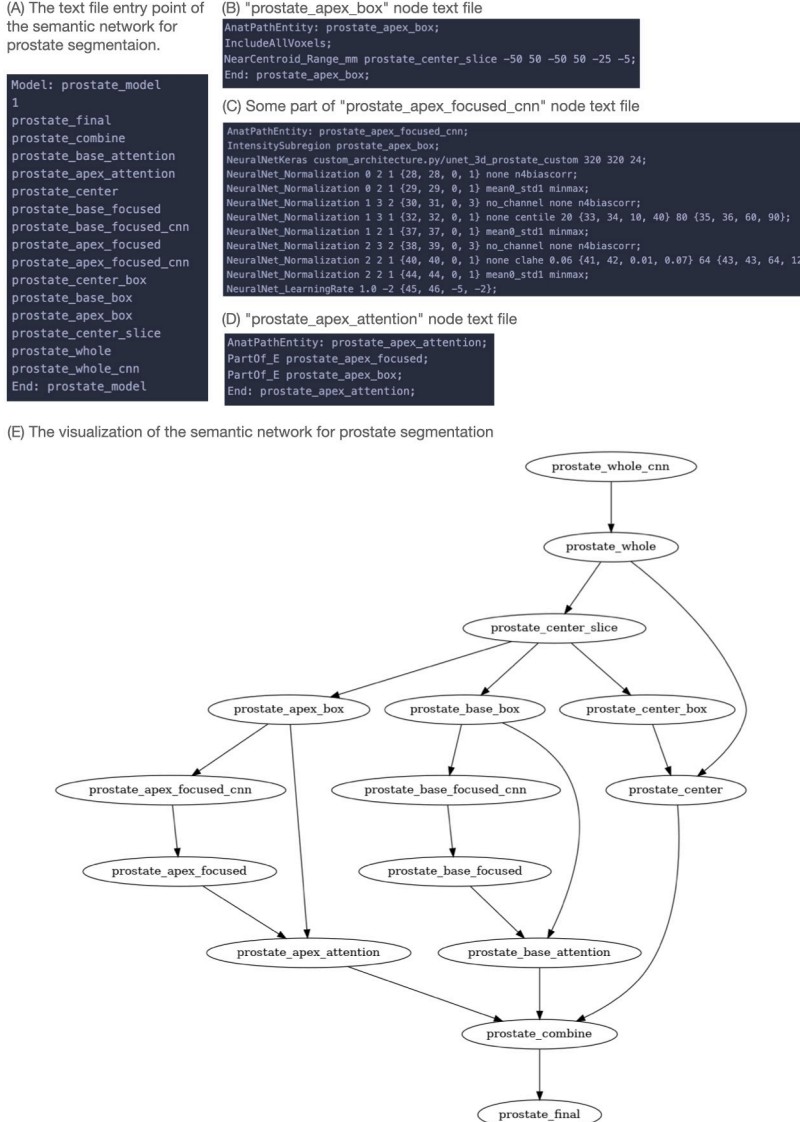

**Fig 4. The SimpleMind semantic network (SN) for prostate segmentation.** The SN of a Node List file listing the nodes in the SN and the corresponding set of human readable node files. (A) The prostate SN Node List file; (B) The `prostate_apex_box` node file; The node file is derived with the attributes for recognizing the apex area slices from the initial prediction from the `prostate_whole_cnn` node. (C) The `prostate_apex_focused_cnn` node file. (D) The `prostate_apex_attention` node file. (E) Graph visualization of the prostate SN.

The `prostate_apex_box` node in Fig 4B describes a box-shaped region that contains the prostate apex:

- Line 3 describes a box that is 100 x 100 mm in plane and 5 to 25 mm above the center slice of the prostate, which is in turn derived from the initial approximate segmentation provided by the `prostate_whole_cnn` node

The `prostate_apex_focused_cnn` node trains a DNN specifically focused on the prostate apex using the following attributes:

- Line 2 bases input image normalizations on the local histogram of the *prostate_apex_box*, mimicking the region-specific window/leveling (contrast enhancement) observed with human experts

- Lines 4–11 specify input image normalization methods with tunable parameters that make use of the local histogram

- Line 12 specifies the tunable learning rate parameter for training DNN weights.

The *prostate_apex_focused_cnn* node has additional attributes not shown in the figure for DNN learning schemes, such as batch size, DNN weight optimizer, etc. These parameters are refined and optimized for the apex region as the system learns from data (see Section Learning and Optimization).

The *prostate_apex_attention* node describes the attributes of the prostate apex region:

- Line 3 indicates that it is within the *prostate_apex_box*.

- Line 4 indicates that it is further part of the *prostate_apex_focused*, which is the smoothed output of the *prostate_apex_focused_cnn*.

These attributes demonstrate spatial reasoning between different objects to refine and cross-check regions.

Similar nodes were created for the prostate base region to add similar thinking to the *prostate_base_focused_cnn* node. Then, the prostate SN combines regions from the apex/base- and center-focused predictions just as the human expert seamlessly integrates their visual processing of each region.

We compare the SimpleMind prostate application with a single 3D U-Net as a baseline, the same DNN model used in *prostate_whole_cnn*. The DNN models are trained with the 40 images from the PROMISE12 (Prostate MR Image Segmentation 2012 Challenge) dataset [14], publicly available in the Grand Challenge (https://promise12.grand-challenge.org). The remaining 10 images are used for the evaluation. The results are fully reproducible with the public SimpleMind prostate application (https://gitlab.com/sm-ai-team/simplemind-applications/-/tree/develop/mri_promise12_quick_start), and any information, including the DNN training details, are fully specified in the knowledge base. Figs 5 and 6 show results from the SimpleMind prostate application. Fig 5 demonstrates SimpleMind adding thinking to the preprocessing of DNN inputs. Fig 5A shows an original axial MR image slice, Fig 5B shows the segmentation of the *prostate_apex_box* node as a red overlay on a coronal view, and Fig 5C shows the preprocessed DNN input image used by the *prostate_apex_focused_cnn* node, i.e., normalized based on the local histogram of the *prostate_apex_box*. Normalizing based on the local histogram of the *prostate_apex_box* node enhances contrast between the prostate apex and background. For comparison, Fig 5D shows the normalized input image for the *prostate_whole_cnn* node having much less contrast. Thus SimpleMind is able to describe and make use of expert human knowledge akin to choosing a window/level setting specific to the prostate apex that is different than for the central prostate region.

Fig 6 shows an example of prostate segmentation by SimpleMind and from the baseline single DNN model. Fig 6A shows the original axial MR image slice and Fig 6B the reference prostate segmentation overlay with a green contour. Fig 6C shows the *prostate_whole_cnn* segmentation (the baseline result) that includes a dumb mistake by the DNN in including a discontiguous region as part of the prostate. When recognizing the *prostate_whole* node, a SimpleMind reasoning agent uses the *prostate_whole_cnn* segmentation results and selects a candidate region that best matches the expected size and location attributes, yielding

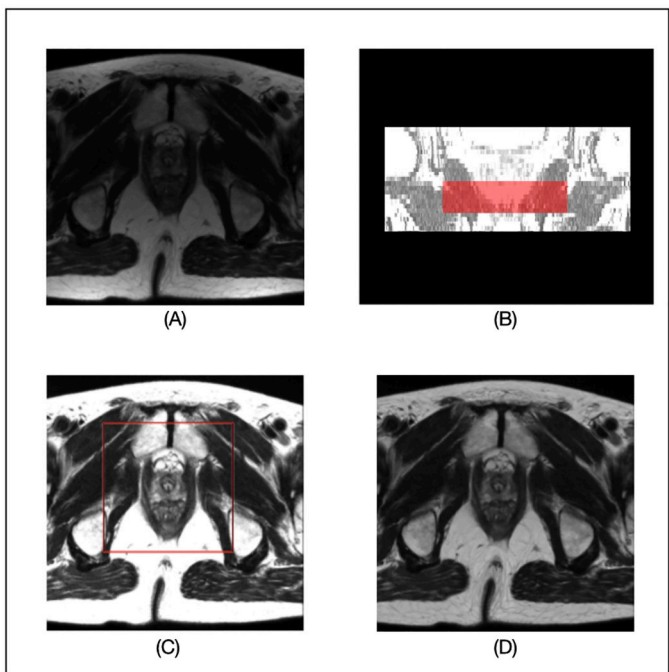

**Fig 5. Preprocessing of DNN inputs for the prostate segmentation.** (A) original image; (B) The
segmented `prostate_apex_box node`; (C) preprocessed image for input to the DNN in the
`prostate_apex_focused_cnn` node (red box indicates the guided area for the local histogram generated from
the `prostate_apex_box node`); (D) preprocessed image used for the DNN in the `prostate_whole_cnn`
node.

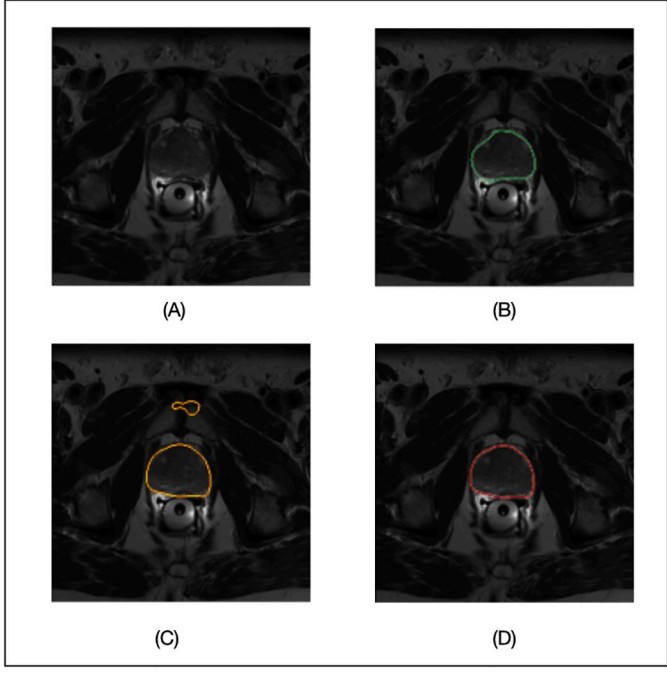

**Fig 6. Prostate segmentation on MRI.** (A) original image; (B) prostate reference segmentation (green contour); (C)
segmentation of the `prostate_whole_cnn` node (the baseline results; yellow contour); (D) SimpleMind
segmentation of the `prostate_whole node`.

the final prostate segmentation result shown in Fig 6D and demonstrating the ability of Simple-Mind to think about DNN outputs. In a preliminary evaluation on 10 MRIs, a baseline single DNN achieved a prostate segmentation Dice Coefficient of 81.8% (`prostate_whole_cnn`). Using the SimpleMind knowledge base combining multiple DNNs, the overall prostate segmentation Dice Coefficient improved to 84.2%, with central, apex, and base regions have coefficients of 90.7%, 88.2%, and 74.1%, respectively.

### Endotracheal tube assessment on chest x-ray

Endotracheal tubes (ETTs) are used to maintain airway patency and lung ventilation. They must be placed correctly to avoid severe complications or even death. Chest x-rays (CXRs) are taken frequently in the intensive care unit to check tube positioning. The medical literature states that the tip of the endotracheal tube should be 5 ± 2 cm above the carina, where the trachea bifurcates into the two main stem bronchi.

Accurate ETT segmentation is difficult due to its narrow diameter, low contrast, and multitude of overlying edges in the CXR image. Further challenges for a DNN alone include: (1) acquiring sufficient examples of misplaced tubes to train and test the system (it occurs infrequently in practice), (2) lack of explainability of how the DNN determines the tube is misplaced, (3) the high level of performance and reliability required in detecting not only the ETT but also the anatomic landmarks for checking its position, given that an error can be fatal. SimpleMind tackles these issues using a knowledge base that describes not only the ETT but also the anatomic landmarks and includes relational attributes to cross-check multiple DNNs and ensure consistency and overall reliability of the system. Rather than attempting to learn misplacement of the tip indirectly from examples, the SimpleMind knowledge base can directly describe when an alert should be given based on the tip location relative to the carina.

The semantic network shown in Fig 7 includes DNNs for the trachea (`trachea_cnn`), carina (`carina_cnn`), and ETT (`et_tube_1_cnn` and `et_tube_2_cnn`). It defines a "safe zone" for the ETT tip using spatial concepts:

- part of the trachea: Line 3 of the `et_zone_1` node (Fig 7B)

- 3–7 cm above the carina: Line 4 of the `et_zone_1` node—based on the y-coordinate of the centroid (Fig 7B)

- ETT tip must be inside the safe zone: `et_tip_correct` node describes this relative to the `et_zone` node and represents the state of the ETT tip (Fig 7C)

- the ETT path must be within the trachea (and thus not going into the esophagus): `et_path_incorrect` node describes this relative to the `trachea` node and represents the state of the ETT path

- for the ETT position to be correct the two criteria above must be met: this requirement is defined in the `et_tube_correct` node which represents the final decision of the system based on its machine reasoning (Fig 7D)

The model also demonstrates checking of DNN outputs for consistency. The `carina_cnn` outputs a coordinate for the position of the carina, represented by the `carina_1` node. The carina location can also be derived from the inferior portion of the trachea where it branches into the two main stem bronchi, represented by the `carina_2` node. The `carina_3` node indicates that these two should correspond and refines the final result. Accurate detection the

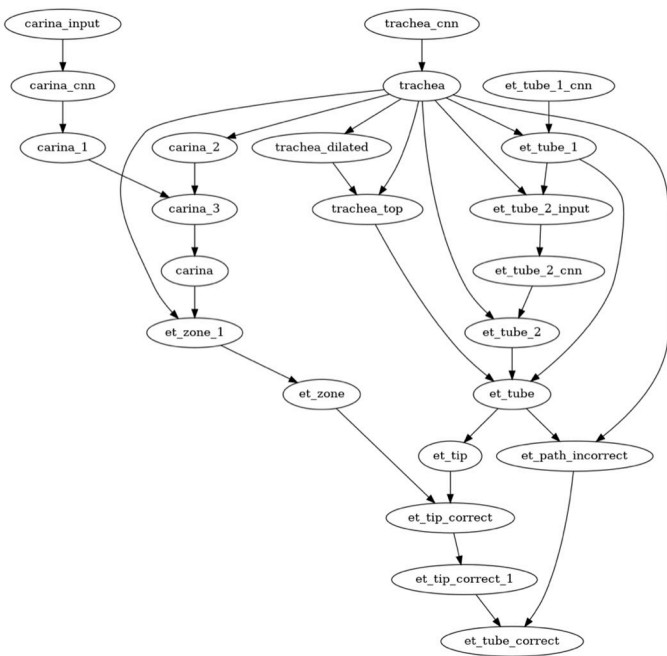

**Fig 7. The SimpleMind semantic network (SN) for the ETT.** The SN of a Node List file listing the nodes in the SN and the corresponding set of human readable node files. (A) The endotracheal tube SN Node List file. (B) The *et_zone_1* node file. (C) The *et_tip_correct* node file. (D) The *et_tube_correct* node file. (E) Graph visualization of the endotracheal tube SN.

the carina is necessary for ETT position checking. Crucially, if the alternate carina locations do not correspond, then the system will report that it is unable to reliably identify the carina rather than outputting an incorrect result. Using knowledge to identify interpretation errors is an important benefit of machine reasoning that allows the system to determine when it is likely to be wrong rather than failing silently.

Figs 8–10 show results from the SimpleMind chest x-ray ETT application. For learning within the knowledge base, we used 2000 images collected retrospectively from ICU patients

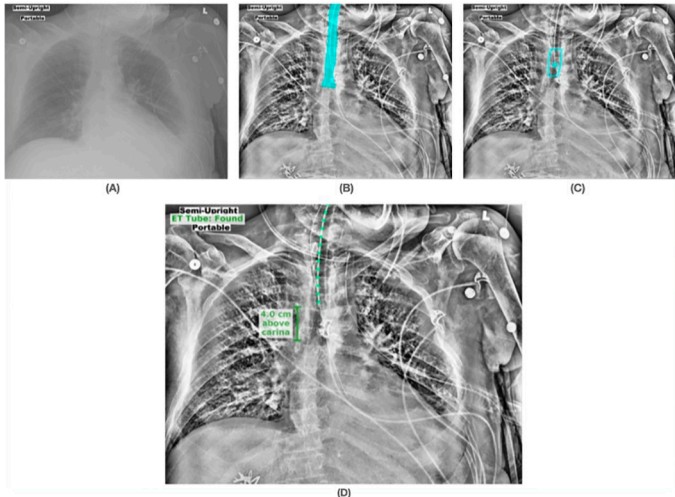

**Fig 8. Visualizations of the nodes involved in the knowledge-supported machine reasoning to determine correct ETT placement.** (A) Original chest x-ray image; (B) trachea region overlayed on an enhanced image; (C) ETT safe zone (*et_zone*) and tube tip location (*et_tip*) showing the tip within the safe zone; (D) the final output of the system as presented to the physician with the green overlay indicating the system determination of correct tube placement.

between April 2018 and September 2019 [5]. All of the DNN nodes are trained with 1488 images, and the application was evaluated with 512 images. These images are restricted for sharing due to patient privacy. Fig 8A shows an original chest x-ray image, Fig 8B shows the *trachea* region overlayed on an enhanced image, Fig 8C shows the ETT safe zone (*et_zone*) and tube tip location (*et_tip*), and Fig 8D is the final output of the system as presented to the ICU physician. The enhanced image is preprocessed specifically for the DNNs; however, human observers also indicated that they find the image useful and hence the AI output is presented as an overlay on this image. Fig 8C shows that the system identified the

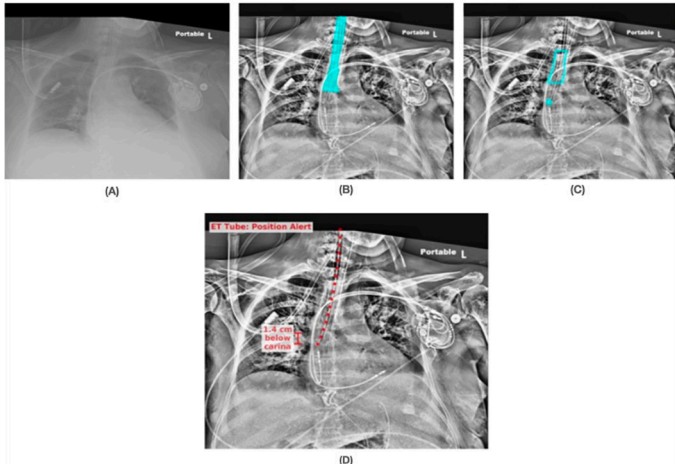

**Fig 9. Visual example of nodes leading to determining correct ETT placement, determined by knowledge-supported machine reasoning.** (A) Original chest x-ray image; (B) trachea region overlayed on an enhanced image; (C) ETT safe zone (*et_zone*) and tube tip location (*et_tip*) showing the tip outside the safe zone; (D) the final output of the system as presented to the physician with the red overlay indicating the system determination of incorrect tube placement (tip too low relative to the carina).

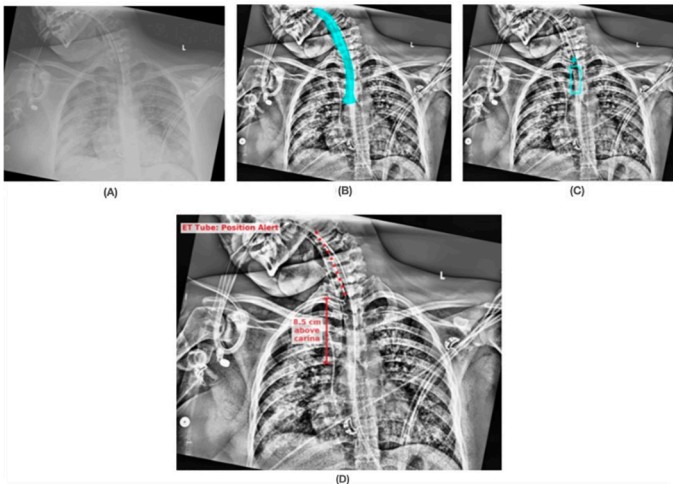

**Fig 10. Visual example of nodes leading to determining incorrect ETT placement, determined by knowledge-supported machine reasoning.** (A) Original chest x-ray image; (B) trachea region overlaid on an enhanced image; (C) ETT safe zone (`et_zone`) and tube tip location (`et_tip`) showing the tip outside the safe zone; (D) the final output of the system as presented to the physician with the red overlay indicating the system determination of incorrect tube placement (tip too high relative to the carina).

ETT tip as within the safe zone, explaining why it thinks the position is correct, and the green color coding of the overlay in Fig 8D indicates to the physician that the AI has identified the tube as correctly placed. Fig 9 shows an example of incorrect tube placement with the red overlay output indicating an alert. Fig 9C shows the ETT tube tip outside the safe zone (tip too low relative to the carina). Fig 10 also shows incorrect tube placement with the tube tip too high relative to the carina (Fig 10C).

The application was evaluated on 512 test CXR images, of which 285 had an ETT present [5]. The AI ETT detection and segmentation was accurate in 89% of cases (454/512). The alert messages indicating that either the ETT was misplaced or not detected had a PPV of 83% (265/320) and NPV of 98% (188/192) [5].

## Learning and optimization

A SimpleMind knowledge base can have a number of attribute parameters that determine the behavior of multiple processing agents. Intelligent behavior emerges from the interaction between agents, and parameters must be co-optimized for them to work in concert. A change in one parameter will alter the optimal setting for others within a node. Also, a SimpleMind application usually links multiple DNNs and their attribute parameters should be co-optimized since the output of one often drives the input of another. An exhaustive search for the optimal parameter combination is beyond the capacity of hand-tuning by a data scientist. In deep learning, hyper parameter tuning is usually performed manually, guided by prior experience, limited and isolated experimentation, and traditional best practices. Using hand tuning, a very small subset of the possible parameter combinations are evaluated. Even using existing AutoML tools, more emphasis is usually placed on learning hyperparameters than parameters implicit in the pre/post processing code that may nonetheless significantly influence performance. Also, existing AutoML implementations typically optimize a single DNN in isolation rather than co-optimizing multiple DNNs [15]. Allowing the co-optimization of all nodes in the knowledge (semantic) network is necessary since nodes are inter-dependent by virtue of

search area dependencies and relational attributes that ensure consistency and allow cross-checking between nodes. For example, if new parameter settings in one node increase or decrease segmentation accuracy, then the parameter describing its spatial relationship to another node may need to be correspondingly updated to a more narrow or larger possible range, respectively.

The SimpleMind environment includes a Knowledge Network Learning and Optimization (KNoLO) method to assess and optimize all attribute parameters in the knowledge base and can encompass AutoML functions. It comprehensively co-optimizes all attribute parameters from all nodes simultaneously, including:

- Object expected characteristics (anatomical knowledge),

- DNN input channels and image preprocessing options,

- DNN learning hyper parameters,

- Post-processing parameters (operating on DNN outputs).

In SimpleMind optimization, a parameter set is a specific combination of values of the tunable attribute parameters in the knowledge base. With a parameter set, the knowledge base becomes fully specified and can be applied to a KNoLO tuning set, yielding a performance for the parameter set. Optimization involves evaluating multiple parameter sets and selecting the best to achieve a data-optimized (tuned) knowledge base.

In the SimpleMind environment, KNoLO is implemented using two primary submodules, an "optimizer" and a "distributor". The optimizer iteratively determines parameter sets to test, which are spawned into jobs by the distributor to compute their performance over the KNoLO tuning set. The distributor can be configured to process jobs sequentially or in parallel. The results are relayed back to the optimizer that then decides which parameter sets to evaluate next. The SimpleMind optimizer and distributor modules can be customized or replaced specific to user needs, preferences, and computing environment. If a parameter set includes tunable DNN learning parameters, SimpleMind will trigger learning of DNN weights. For large-scale optimization involving many parameter sets, parallelization is necessary and available in SimpleMind.

SimpleMind currently provides the genetic algorithm (GA) as a KNoLO optimizer. The GA is an optimization approach inspired by evolution in nature [16]. It encodes all tunable attribute parameters into a "chromosome" where each parameter can be considered a gene that changes system behavior and performance. Each unique chromosome encodes a different parameter set. A group of chromosomes (a "population") represents a group of parameter sets, and each individual chromosome will have a performance value ("fitness score") when evaluated on the KNoLO tuning set. Example fitness functions include dice coefficient (in segmentation tasks), confusion matrix metrics (sensitivity, specificity, precision, recall, etc.—for classification and detection tasks), and distance metrics (e.g. mean squared error—for point detection tasks). Also fitness functions may be derived as a weighted combination of these or other implemented metrics. Based on their fitness, high-performing chromosomes (with the highest fitness scores) are selected to persist to the next iteration ("generation"). Genetic operations, like mutation of chromosome values and crossover between pairs of chromosomes, introduce variation in the population and thus exploration of parameter sets during optimization. As combinations of strong genes propagate and synergize with other complementary genes, the genetic pool evolves to improve performance as defined by the fitness function. While the optimization result via the GA can be affected by randomness, user-defined ranges of tunable parameter sets constrain the search, and reproducibility can be ensured by setting

**Fig 11. The SN node for `prostate_apex_focused_cnn`.** The `prostate_apex_focused_cnn` semantic network node showing parameters exposed for optimization relating to DNN input normalization and learning.

the algorithm seed point consistently. The GA offers an explainable, intuitive approach towards optimization, and has been shown to derive innovative solutions in complex systems in nature, and can handle non-smooth functions better than gradient-based methods [16]. The GA also allows for easy user intervention to guide evolution.

## KNoLO in MR prostate segmentation

As described in Section Prostate Segmentation in 3D MRI, prostate segmentation can be improved by splitting into three sub-tasks: segmentation of the whole-prostate, the base of the prostate, and the apex of the prostate. The attribute parameters of each DNN, including the number of input channels, the preprocessing method and parameters, and the DNN learning hyper parameters (e.g. learning rate), can be exposed for tuning by SimpleMind's KNoLO method. This is demonstrated in the SN node for `prostate_apex_focused_cnn` as shown in Fig 11.

Examining the *Learning Rate* DNN hyper parameter, the default value is explicitly defined as:

`NeuralNet_LearningRate 1.0 -2`

which translates to the default value being set as $1.0 * 10^{-2}$. We can expose this parameter to be tuned by appending a "value-encoding" operator following the default value, for example,

`NeuralNet_LearningRate 1.0 -2 {45, 46, -5, -2}`

This value-encoding operator follows the format,

`{encoded_bit_start_position, encoded_bit_end_position, lower_bound_value, upper_bound_value}`

This operator encodes the possible values to be evaluated by the optimizer as alternatives to the default value of "-2". The value is encoded in the 45th and 46th bits in the binary chromosome of the genetic algorithm, with 2 bits encoding 4 possible values. The lower bound for the 4 possible values is -5, and the upper bound is -2. The rest of the values are equidistant between the lower and upper bounds, resulting in the set of possible learning rate exponents to be $\{-5, -4, -3, -2\}$, corresponding to the chromosome bits, $\{00, 01, 10, 11\}$, respectively.

The `NeuralNet_Normalization` attributes include similar parameter value encoding operators as shown in Fig 11. These parameters control the number of input channels for the DNN model, as well as the preprocessing methods applied to those activated channels and their associated parameters. For instance, bias field correction is a preprocessing step that can be optionally applied to each channel during KNoLO. In this example, the DNN can have up to 3 input channels, with bias field correction, normalization, centile-based preprocessing, and clipping and histogram equalization, as the potential preprocessing steps.

For each of the three neural network nodes (whole, base, apex) in the prostate SN, this set of tunable parameters (preprocessing method, number of channels, and learning rate) were exposed for tuning. Through 10 iterations (generations with population size 30) within

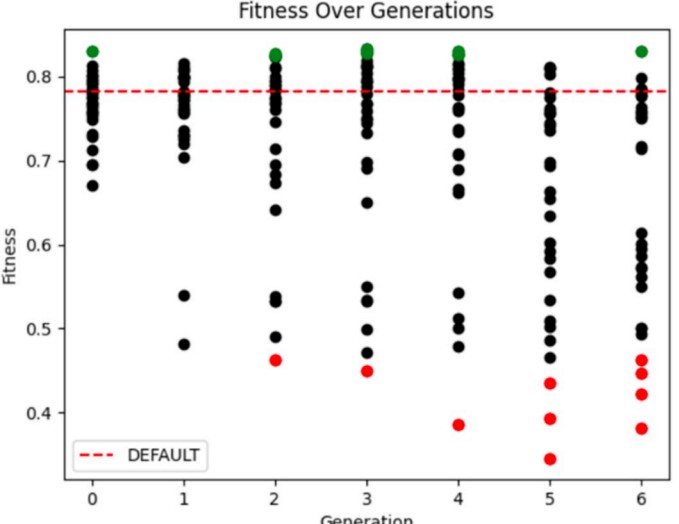

**Fig 12. Evolution of fitness performance of chromosomes (i.e., different parameter sets) over time within the GA optimization in prostate segmentation.** The y-axis represents the fitness score for each chromosome in a population, representing the weighted average of the prostate segmentation performance of the apex, base, and whole prostate regions, averaged across the cases of the KNoLO tuning set. For comparison, the red dotted line demarcates the performance of the prostate segmentation SN using only hand-tuned parameters by a data scientist. Each generation (iteration) has 30 chromosomes, with the green points representing the highest fitness chromosomes and the red points representing the lower fitness chromosomes. The highest fitness score was 0.833.

KNoLO, approximately 300 different parameter sets were tested on a tuning set of 10 cases (with the DNNs trained on a separate 30 cases). Fig 12 visualizes the optimization by the genetic algorithm with fitness scores for generations 3 through 10. For comparison, the red dotted line demarcates the performance of the prostate segmentation SN using only hand-tuned parameters by a data scientist, which was exceeded by the auto KNoLO optimization.

It was found that within the first 10 generations, the best performing parameter set included parameters that set the various preprocessing steps with multiple input channels across the DNN nodes to be a combination of bias-field correction, min-max normalization, and clipping and histogram equalization. (see Table 1). Fig 13 displays the images and pixel intensity histograms before and after these preprocessing steps. After GA optimization, the *prostate_whole_cnn* had two channels, one using bias field correction and

**Table 1. Image preprocessing steps selected by the KNoLO optimizer for each DNN input channel for the three prostate regions.**

|  | Channel 1 | Channel 2 |
|---|---|---|
| *prostate_whole_cnn* | Bias Field Correction | Bias Field Correction, |
|  | Min-Max Normalization | Clipping & Histogram Equalization |
| *prostate_apex_focused_cnn* | Bias Field Correction | Bias Field Correction |
|  | Min-Max Normalization | Min-Max Normalization |
| *prostate_base_focused_cnn* | Bias Field Correction | - |
|  | Min-Max Normalization | |

After optimization via KNoLO, different combinations of preprocessing steps for input channels were found to be optimal for each of the CNNs focused on different prostate regions.

## Whole Prostate CNN

**Bias Field Correction**
**Min-Max Normalization**

Channel 1

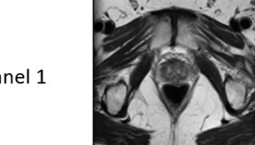 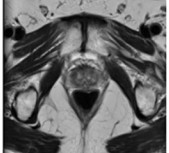 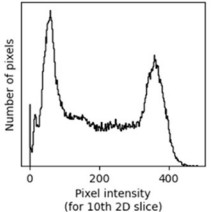

10th 2D slice [original image]   10th 2D slice [preprocessed image]

Channel 2

**Bias Field Correction**
**Clipping & Histogram Equalization Normalization**

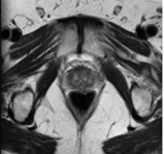 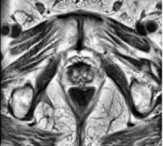 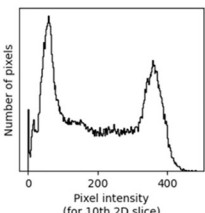

10th 2D slice [original image]   10th 2D slice [preprocessed image]

## Apex CNN

**Bias Field Correction**
**Min-Max Normalization**

Channel 1

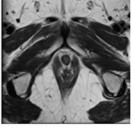 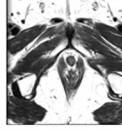 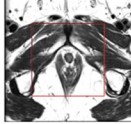 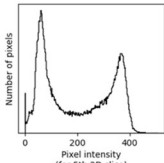 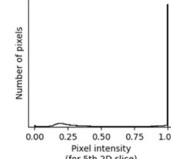

5th 2D slice [original image]   5th 2D slice [preprocessed image]   5th 2D slice ph area

Channel 2

**Bias Field Correction**
**Min-Max Normalization**

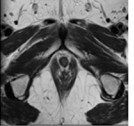 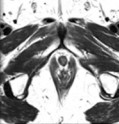 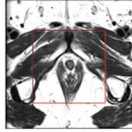 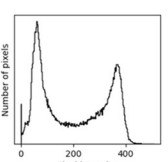 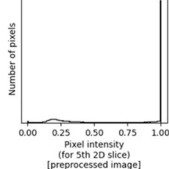

5th 2D slice [original image]   5th 2D slice [preprocessed image]   5th 2D slice ph area

## Base CNN

**Min-Max Normalization**

Channel 1

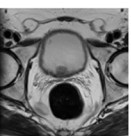 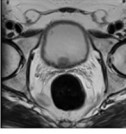 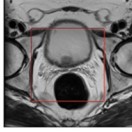 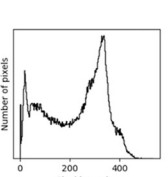 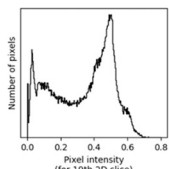

19th 2D slice [original image]   19th 2D slice [preprocessed image]   19th 2D slice ph area

**Fig 13. Example 2D image slices and their associated pixel intensity histograms before and after preprocessing.**
Through KNoLO, an optimal combination of preprocessing steps and input channels was found for each prostate region specific CNN, resulting in the displayed example slices.

min-max normalization; the *prostate_apex_focused_cnn* used two channels with bias field correction filters followed by min-max normalization; and the *prostate_base_focused_cnn* used one channel with min-max normalization applied as preprocessing. For both the apex and base DNNs, preprocessing was done based on a focused, localized region immediately surrounding the prostate region (derived from the *prostate_whole_cnn*), to provide normalization and preprocessing more specific to the local area of the prostate. The optimal learning rates for these input channels were, 0.01 (*prostate_whole_cnn*), 0.0001 (*prostate_apex_focused_cnn*), and 0.001 (*prostate_base_focused_cnn*). Together, this led to a fitness of 0.833, performing with an average dice coefficient score of 0.842 in segmenting the overall prostate across the KNoLO tuning set—and surpassing the hand-tuned fitness score of 0.78. It is significant to note that the learning rates were co-optimized with the image preprocessing; different input images may require different learning hyperparameters for optimal performance. The whole KNoLO process is reproducible for the public SimpleMind prostate application (https://gitlab.com/sm-ai-team/simplemind-applications/-/tree/develop/mri_promise12_quick_start) using the PROMISE12 dataset (https://promise12.grand-challenge.org).

## SimpleMind software environment

SimpleMind is an open-source Cognitive AI software environment that enables users to create SimpleMind applications for image understanding. Application development involves creating a knowledge base in the human-readable language of SimpleMind and then tuning its parameters using the Learn module. At runtime, the knowledge base is applied by the Think module to process an image and recognize objects. There is a quick-start tool for a user to build a new SimpleMind application software from a given knowledge base and configurations. The environment can be extended through application programming interfaces (APIs) whereby developers can implement new processing algorithms as agents or add optimization techniques.

**Knowledge Base** A user creates a knowledge base, in the form of a semantic network (SN), as structured text files following the SimpleMind language. When creating a semantic network, **SN Summary Tools** aid users by providing (1) a visualization of the relational links between nodes (Figs 2E, 4E and 7E were generated by this tool); (2) a text summary of the chromosomes and exposed parameters tunable with the Learn module.

**Think Module** The Think module implements multi-agent thinking (see Section Multi-Agent Thinking) to execute the image understanding of a SimpleMind application. Inputs to the Think module are:

- a knowledge base

- an input image

- (optionally) a specific set of knowledge base attribute parameters (a parameter set i.e. "chromosome")

The Think Module reads inputs, builds a Blackboard from the SN (using the Knowledge Mapping agent), iteratively runs Scheduling, Segmentation, and Reasoning agents, and writes the Blackboard and segmented image regions to output files. Thus the outputs of the Think module are:

- a Blackboard output file instantiated to the input image with each Solution Element containing image candidates, features values, selected best candidate for the object it represents

- individual files containing the image regions selected as the best match to each object

The environment provides a **Blackboard Viewer** to visualize the results contained in each Solution Element as overlays on the input image. It can show all candidate regions and the selected best candidate. A **Blackboard Summary Tool** can be used when testing on a set of images. It generates a summary HTML report of the multiple blackboards from the set of images, including key images with segmentation results overlaid.

**Learn Module** The Learn module implements the KNoLO method, including tuning of all SN parameters and training of DNN weights. The inputs to the Learn module are:

- the knowledge base to be optimized

- the KNoLO tuning set of cases to which optimization will be performed

- (optionally) training sets for DNN nodes for learning weights

The KNoLO process will search for the optimal parameter set (chromosome) for the knowledge base on the given tuning set. The Learn module outputs:

- the best performing chromosome

- the chromosome-specified (optimized) knowledge base

During optimization, the Learn module will call the Think module for each parameter set of interest. This generates the following for each parameter set and each input image in the KNoLO tuning set:

- a knowledge base with a specified parameters

- an instantiated Blackboard of Solution Elements with best matched image regions

Users can access these outputs during the optimization process. As seen in the prostate KNoLO example, KNoLO processing can involve the computation of many hundreds (or sometimes thousands) of parameter sets, and processing each parameter set may involve training and executing multiple DNNs. For optimization of a knowledge base to be feasible, parallelization of processing is necessary. This is enabled via the KNoLO Distributor, with current support for HTCondor [17] and further support for other job distribution systems (Airflow, Kubernetes) in development. When automatically tuning parameters, a **KNoLO Status Summary Tool** provides an analysis of chromosome performance in csv and graphs in HTML report format. The KNoLO Status Summary tool helps users track the overall progress of automatic parameter tuning within the Learn module.

## Open source implementation

The open-source SimpleMind environment is hosted on the public GitLab repository (https://gitlab.com/sm-ai-team/simplemind). It supports Docker installation with the official Docker image from the public DockerHub repository. The official documentation (https://sm-ai-team.gitlab.io/simplemind/sm.html) includes the detailed information and examples about the input configurations and the step-by-step explanations about how to install the environment, how to build an application using the environment, and how to teach and execute the applications. Various example applications are provided in https://gitlab.com/sm-ai-team/simplemind-applications, including links to public data sets for provided kidney and prostate segmentation application examples.

**Compatibility** The SimpleMind environment is compatible with Linux operating system only. Experimental results were calculated on Intel(R) Xeon(R) Gold 5220R CPU @ 2.20GHz.

**Dependencies** The environment integrates existing open-source libraries within agents in the Think/Learn Modules (e.g., the DNN agent in the Think Module uses Tensorflow) and is

open for users to incorporate other open-source libraries (e.g. PyTorch, NVIDIA Clara, nnU-Net) or custom algorithms. All dependencies are wrapped and supported via the Docker environment. In the Docker environment, CUDA version 10.0, Python version 3.7.2, Tensorflow version 1.15.0, Keras version 2.3.1 were used. The remainder of SimpleMind's dependencies can be found in the Dockerfile within the software package repository.

## Discussion

Cognitive AI is broadly defined as enabling human level reasoning and intelligence and more recently there is an emerging interest in the area of "neuro-symbolic" AI that seeks to bring together neural and symbolic approaches in a best-of-both-worlds scenario. A recent survey article identified a small but increasing number of publications at top tier AI conferences [18]. However, there have been few general-purpose implementations and computing architectures for neuro-symbolic AI and for computer vision specifically.

Neural-symbolic research starts from two observations: (1) the physical implementation of our mind is based on the neural system; (2) abstract thinking is based on symbol manipulation and complex symbolic data structures (like graphs, trees, shapes, and grammars) [19]. However, it is unclear at present how symbolic processing emerges from neural activity, and at present there exists no viable alternative to symbolic approaches in order to encode complex tasks. A possible response is to break the modeling problem into two steps: the first, a symbolic modeling of cognitive behavior and the second, a mapping from symbolic to artificial neural network [19]. SimpleMind achieves this by embedding deep neural networks within a symbolic knowledge base. It can be considered a "hybrid learning system" that brings together features from connectionism (e.g., DNNs) and symbolic AI. Four key advantages have been suggested as arising from this combined approach [18]: (1) interpretability, (2) error recovery, (3) out of distribution (OOD) handling, and (4) learning from small data. We will describe how SimpleMind supports and improves DNNs in terms of these four advantages of neuro-symbolic AI.

### Supporting and improving deep neural networks with SimpleMind reasoning

While there have been encouraging results from deep neural networks in controlled experiments, there is a notable lack of translation into routine use in radiology clinical practice. SimpleMind supports DNNs with machine reasoning to increase real-world reliability and explainability of decisions. The neural and semantic networks store, represent, and use knowledge in different ways and are complementary. The embedding of DNNs in a SimpleMind semantic network confers three primary benefits:

1. It allows explicit knowledge to be applied systematically during pre and post processing, including the following to improve performance and particularly reliability.

- Computing a search area in which to apply DNN segmentation using spatial relationships in the semantic network.

  - This has potential utility in approaches based on guiding attention [20, 21].

- Invoking additional segmentation agents to refine the DNN segmentation result, e.g., using morphological operators etc.

- Selection of the best candidate image region outputted by the DNN based on expected characteristics defined in the knowledge base, or conversely, rejection of the output if it does not meet expectations.

  - Thus SimpleMind enables reasoning on multiple detected objects to ensure consistency, providing cross-checking between DNN outputs. The more of a scaffold of knowledge that exists, the more robust the overall image interpretation can be. Furthermore, it has been recognized that central to construction of a "mental model" is the formation of a combined whole representation from sets of part representations and that establishing global coherency is a key aspect of human reasoning [19].

  - The ability of SimpleMind to reject candidates for an object, and output nothing, is in contrast to DNNs that always output a result and can fail silently. It does not preclude recognition of subsequent objects based on other knowledge and avoids propagating errors. This gives SimpleMind applications more resilience in handling OOD cases and error recovery.

2. It provides a high degree of interpretability and explainability, deriving from the knowledge base and the Blackboard.

  - The knowledge base makes explicit the knowledge that was previously implicit in pre and post processing code and makes it easier to apply more knowledge intuitively.

    - The knowledge base and the initialized Blackboard make the object recognition model highly interpretable. Also, by exposing parameters in the knowledge base the result of optimization is transparent and a human can oversee the process and discover which parameters have the greatest impact on performance.

    - The human also has the ability to override the optimization of any parameter, for example, if it appears to be overfit to a data set or if they have an intuition that there is a more optimal value (which is readily evaluated within the SimpleMind framework and further optimized by KNoLO).

  - The thinking of SimpleMind as it processes an image is captured in the Blackboard. A human can know what it was thinking by reviewing the Blackboard contents.

    - The instantiated Blackboard provides a high degree of explainability as to how SimpleMind reached its decisions about the image. For example, the image search area for a given object can be interpreted by reviewing the spatial relationships to other objects on the Blackboard. This allows a user to understand how the search area was derived and detect knowledge assumptions that might be wrong or improperly specified.

3. Using a human-provided knowledge base, SimpleMind can perform object recognition with little or no training data.

  - When there is insufficient data to train a DNN, other segmentation agents (e.g., intensity thresholding or edge detection) can use the knowledge base to generate initial segmentation results. Little or no training data is needed since the initial semantic network can be constructed using declarative knowledge engineering rather than machine learning. These initial results can be used with manual editing to generate the necessary training sets for DNN learning.

- Thus SimpleMind strongly supports learning from small data. When an OOD situation arises it can be added to the knowledge base and handled without training data being initially available.

These benefits are also consistent with goals of trustworthy AI according to the High-Level Expert Group on AI from the European Commission [22], in particular the following guidelines:

- Transparency: AI systems and their decisions should be explained.

- Technical Robustness and safety: AI systems need to be resilient with a fall back plan in case something goes wrong.

- Human agency and oversight: AI systems should empower human beings, allowing them to make informed decisions with proper oversight mechanisms.

A driver for neural-symbolic research has been stated as the "aim to create a new, neurally-inspired computational platform for symbolic computation which would, among others, be massively distributed with no central control, robust to unit failures, and self-improving with time" [19]. SimpleMind can be seen as a step in this direction as it allows knowledge and agents to be added without modification of its simple control system, its semantic network adds robustness to failures in the recognition of any single node, and it provides the KNoLO method for self-improvement.

## SimpleMind as a scalable virtual data scientist

The Cognitive AI implemented in SimpleMind uses a knowledge base to chain together multiple DNNs and pre/post processing algorithms and perform machine reasoning with their outputs. It also includes knowledge base learning and optimization, encompassing AutoML. A vision system typically involves the following modules: (1) image preprocessing, (2) DNN segmentation, (3) segmentation post-processing. SimpleMind allows for a knowledge-based description to drive (1) and (3), rather than ad hoc implicit use of knowledge in computer code, and provides a rich and expandable knowledge base of methods with parameter optimization. The result is a more flexible and scalable environment.

The knowledge base is independent of image processing algorithms, with a descriptive form that is intuitive, understandable, and able to be created, expanded, and maintained, by domain experts (rather than necessarily requiring a data scientist). This allows the expert knowledge to be reused and applied to other problems, for example, analysis of images acquired by different imaging modalities. In contrast, many applications incorporate domain knowledge by embedding it implicitly within segmentation algorithms. SimpleMind allows a set of general algorithms to be applied to a wide range of tasks by simply extending a high-level descriptive model.

In the SimpleMind environment, agent inputs and outputs must respect Blackboard data structures, but the processing they perform is modular, encapsulated, and highly flexible. Agents maintain a degree of autonomy with their interactions regulated via the Blackboard and its control system. For example, a new segmentation algorithm can be developed and added to SimpleMind without any direct awareness of other agents, how knowledge is represented, or even the problem of medical imaging at all. Parameters used by agents (meta level knowledge) are derived from the knowledge base and transformed to the Blackboard (rather than hard coded implicitly). This means that they are not fixed but can be adapted to a particular problem by the Learn module. This is also liberating for the algorithm developer in that SimpleMind takes care of integrating, tailoring, and optimizing that algorithm for the specific

task. In effect, it is able to generalize the agent skill and apply it as needed to any vision problem, a characteristic of intelligence.

Currently, the work of a data scientist training deep neural networks involves hand tuning of parameters and applying knowledge ad hoc in heuristic pre/post processing algorithms. The result is application-specific Narrow AI, that is typically suboptimal in terms of parameter search and limited as to the level of knowledge and reasoning applied. A computer programmer or data scientist uses domain knowledge when developing an image processing algorithm. However, their expectations for the image content are often reflected implicitly in the processing steps of their software. SimpleMind supports two forms of knowledge for developing AI:

1. Concept knowledge: knowledge of the scene and reasoning about the content (derived from an application expert);

2. Processing knowledge: meta knowledge about learning and image processing (normally provided by a data scientist).

SimpleMind serves as a virtual data scientist in two key ways:

1. It applies domain knowledge during image analysis—to guide segmentation and check results (typically a data scientist does this ad hoc within pre- and post- processing code)

2. It automatically tunes parameters—including deep learning hyper parameters and parameters associated with pre/post processing

   - Hand-tuning of a large number of inter-dependent parameters is infeasible, especially on an ongoing basis as more data is made available through real-world use.

   - SimpleMind co-optimization of all agent parameters provides a more extensive, unbiased search than is possible by a human data scientist.

By incorporating these activities that are typically done by a human data scientist, we achieve a virtual data scientist that is more scalable and better able to optimize than a human. Rather than tacitly applying knowledge ad hoc within coded algorithms, SimpleMind accepts high level descriptions of scene content and concepts and automatically chains together agents for image pre/post processing, DNNs, and machine reasoning. Thus SimpleMind brings transparency, efficiency, scalability, and reliability to AI development and automates tasks of the data scientist.

The SimpleMind environment has improved the performance of a number of medical imaging applications [5, 6, 8], and we believe that there is strong potential utility for broader research and commercial applications in building trustworthy AI. The open source software allows for knowledge base expansion and agent aggregation by a community of developers. SimpleMind aggregates, supports, and improves deep neural networks by embedding them within a Cognitive AI framework.

## Acknowledgments

The authors wish to thank their friends and colleagues at the UCLA Center for Computer Vision and Imaging Biomarkers.

## Author Contributions

**Conceptualization:** Youngwon Choi, M. Wasil Wahi-Anwar, Matthew S. Brown.

**Data curation:** Youngwon Choi, M. Wasil Wahi-Anwar, Matthew S. Brown.

**Formal analysis:** Youngwon Choi, M. Wasil Wahi-Anwar, Matthew S. Brown.

**Investigation:** Youngwon Choi, M. Wasil Wahi-Anwar, Matthew S. Brown.

**Methodology:** Youngwon Choi, M. Wasil Wahi-Anwar, Matthew S. Brown.

**Project administration:** Matthew S. Brown.

**Resources:** Matthew S. Brown.

**Software:** Youngwon Choi, M. Wasil Wahi-Anwar, Matthew S. Brown.

**Supervision:** Matthew S. Brown.

**Validation:** Youngwon Choi, M. Wasil Wahi-Anwar, Matthew S. Brown.

**Visualization:** Youngwon Choi, M. Wasil Wahi-Anwar, Matthew S. Brown.

**Writing – original draft:** Youngwon Choi, M. Wasil Wahi-Anwar, Matthew S. Brown.

**Writing – review & editing:** Youngwon Choi, M. Wasil Wahi-Anwar, Matthew S. Brown.

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
