## [Decision Letter · Decision Letter 0]

10 Jan 2023

PONE-D-22-34289SimpleMind: An open-source software framework that adds thinking to deep neural networksPLOS ONE

Dear Dr. Brown

Thank you for submitting your manuscript to PLOS ONE. Your manuscript has now been reviewed by experts in the field. Please revise the manuscript according to the referees' comments and upload the revised file.

We look forward to receiving your revised manuscript.

Kind regards,

Ayesha Maqbool, PhD

Academic Editor

PLOS ONE

3. We noted in your submission details that a portion of your manuscript may have been presented or published elsewhere. [A preprint of this article was uploaded to the arXiv archive: https://arxiv.org/abs/2212.00951] Please clarify whether this  publication was peer-reviewed and formally published. If this work was previously peer-reviewed and published, in the cover letter please provide the reason that this work does not constitute dual publication and should be included in the current manuscript.

Reviewers' comments:

Reviewer's Responses to Questions

**Comments to the Author**

1. Is the manuscript technically sound, and do the data support the conclusions?

Reviewer #1: Yes

Reviewer #2: Yes

2. Has the statistical analysis been performed appropriately and rigorously? 

Reviewer #1: N/A

Reviewer #2: N/A

3. Have the authors made all data underlying the findings in their manuscript fully available?

Reviewer #1: No

Reviewer #2: Yes

4. Is the manuscript presented in an intelligible fashion and written in standard English?

Reviewer #1: Yes

Reviewer #2: Yes

5. Review Comments to the Author

Reviewer #1: This paper proposes an open-source software framework, i.e., SimpleMind, for explainable AI in medical image analysis. Since the “black-box“ property of deep neural networks greatly hinders its application in medical image analysis, how to enhance the interpretability and build trustworthy AI system is an important topic. Thus the idea and motivation of this work are good. The authors use three experiments with different modalities (i.e., CT, MRI, X-ray) to illustrate the usage of the software, and verify its effectiveness in medical image analysis.

There are several comments, concerns, and suggestions for this work.

[1] Since this is a software paper rather a technical/theoretical paper, it is important to clearly illustrate the overall structure design of this software. The current version is not good enough on this point. What’s the input/output of the software? How many modules and what are their roles?

[2] As a software paper, a detailed description of the workflow of the software is essential. For a reader who is interested in this work, how to use the software step-by-step? Does this software have a user-interface? Adding a section about this content is needed.

[3] Currently, interpretable AI has become a very hot topic. As for DNN, many works such as attention mechanism have greatly enhanced the interpretability of DNNs. What’s the advantage (or differences) of this work over these methods (e.g. attention techniques in DNNs)?

[4] The experiments lack comparisons with some baselines. For example, set a classic DNN (ResNet, UNet, …) as the baseline, then compare the proposed software with it. Showing some failure cases of baseline is helpful to illustrate the effectiveness of the proposed method.

[5] The genetic algorithm (GA) has some randomness. Will this influence the result (output) of this software?

Reviewer #2: The authors developed a software framework called “SimpleMind” for reliable medical image analysis. One extremely interesting aspect of the SimpleMind framework lies in its ability to incorporate and utilize domain knowledge, which is important for analyzing different types of medical images [1]. This paper is very well written, the experiments are comprehensive, and its technical contributions are solid. Thank the authors for this amazing work! I look forward to seeing further extensions and applications of this framework on various medical imaging tasks, like classification and detection in the future.

Just one minor comment: Figure 2, 4, 7, 11, 13 are blurry, making texts the image captions unreadable. I guess this is because the journal’s submission system failed to preserve the original resolutions. Please make sure higher-resolution versions are used for all the figures when it comes to publishing.

[1] "Recent advances and clinical applications of deep learning in medical image analysis." Medical Image Analysis (2022).

6. PLOS authors have the option to publish the peer review history of their article (what does this mean?). If published, this will include your full peer review and any attached files.

Reviewer #1: No

Reviewer #2: No

---

## [Author Response · Author response to Decision Letter 0]

1 Mar 2023

The responses were included separately with a file "Response to Reviewers.pdf".

---

## [Editor Report · Decision Letter 1]

13 Mar 2023

SimpleMind: An open-source software environment that adds thinking to deep neural networks

PONE-D-22-34289R1

Dear Dr. Brown,

We’re pleased to inform you that your manuscript has been judged scientifically suitable for publication and will be formally accepted for publication once it meets all outstanding technical requirements.

Kind regards,

Ayesha Maqbool, PhD

Academic Editor

PLOS ONE
---

## [Editor Report · Acceptance letter]

3 Apr 2023

PONE-D-22-34289R1 

SimpleMind: An open-source software environment that adds thinking to deep neural networks 

Dear Dr. Brown:

I'm pleased to inform you that your manuscript has been deemed suitable for publication in PLOS ONE. Congratulations! Your manuscript is now with our production department. 

Kind regards, 

on behalf of

Dr. Ayesha Maqbool 

Academic Editor

PLOS ONE